# Microeukaryotic Communities of the Long-Term Ice-Covered Freshwater Lakes in the Subarctic Region of Yakutia, Russia

Yuri Galachyants [1,*], Yulia Zakharova [1], Maria Bashenkhaeva [1], Darya Petrova [1], Liubov Kopyrina [2] and Yelena Likhoshway [1]

1 Limnological Institute, Siberian Branch of the Russian Academy of Sciences, 3 Ulan-Batorskaya Street, Irkutsk 664033, Russia
2 Institute for Biological Problems of Cryolithozone, Siberian Branch of the Russian Academy of Sciences, 41 Lenin Ave, Yakutsk 677980, Russia
* Correspondence: yuri.galachyants@lin.irk.ru

**Abstract:** Currently, microeukaryotic communities of the freshwater arctic and subarctic ecosystems are poorly studied. Still, these are of considerable interest due to the species biogeography and autecology as well as global climate change. Here, we used high-throughput 18S rRNA amplicon sequencing to study the microeukaryotic communities of the large subarctic freshwater lakes Labynkyr and Vorota in Yakutia, Russia, during the end of the ice cover period, from April to June. By applying the statistical methods, we coupled the microeukaryotic community structure profiles with available discrete factor variables and hydrophysical, hydrochemical, and environmental parameters. The sub-ice layer and the water column communities were differentiated due to the temporal change in environmental conditions, particularly temperature regime and electric conductivity. Additionally, the community composition of unicellular eukaryotes in lakes Labynkyr and Vorota was changing due to seasonal environmental factors, with these alterations having similar patterns in both sites. We suggest the community developed in the sub-ice layer in April serves as a primer for summer freshwater microeukaryotes. Our results extend the current knowledge on the community composition and seasonal succession of unicellular eukaryotes within subarctic freshwater ecosystems.

**Keywords:** subarctic freshwater lakes; unicellular eukaryotes; seasonal changes; water column; sub-ice layer; high-throughput 18S rRNA amplicon sequencing

## 1. Introduction

Arctic and subarctic freshwater ecosystems are apparently insufficiently studied with respect to microeukaryotic community composition, as well as its spatial variability and temporal dynamics [1]. However, this problem is not only of significant interest with regard to biogeography, autecology and paleohistory of individual species, but is also important for environmental monitoring, for the study of under-ice water ecosystems and in the context of global climate changes [2]. In this respect, two large freshwater subarctic lakes, Labynkyr (LL) and Vorota (LV), which are located in close proximity to the cold pole of Eurasia, are very valuable objects. Until recently, communities of unicellular eukaryotes in these lakes were not analyzed by metagenomics or high-throughput sequencing (HTS). A typical subarctic freshwater ecosystem is characterized by a low temperature, not exceeding 10 °C even in the warmest summer period, decreased concentrations of dissolved inorganic nutrients, low primary production, and oligotrophic state [3,4]. LL and LV are covered by ice for nearly two-thirds of the year; therefore, changes in the community composition between the ice cover and the open water periods are of particular interest. Recent studies of these lakes have shown high diversity and unique composition of microalgae by optical and electron microscopy [5–9] and bacteria by high-throughput sequencing [10].

Extensive knowledge of the under-ice environmental conditions in lake ecosystems has already been accumulated. This might help us to understand how a particular lake

ecosystem will evolve over the next season. The under-ice environmental monitoring and evaluation of seasonal connectivity can improve our understanding of the environmental response to global climate change [11]. However, only a small number of such studies consider microeukaryotic community structure as a proxy for continuous environmental monitoring in freshwater lakes. For instance, the long-term study of phytoplankton in oligotrophic Lake Tovel, Brenta Dolomites, during the ice-covered and open water periods was recently reported [12]. However, investigation of the microeukaryotic communities in oligotrophic mountain, subarctic and arctic lakes is often limited by late spring and summer periods due to the difficulty of accessing the sampling sites in these regions.

Within the ice covered period, the environmental conditions are different between the sub-ice layer adjacent to the bottom ice surface and the upper layer of the water column. Further, conditions in both environments are changing during the ice cover period, which results in continuous differentiation of the under-ice communities of microeukaryotes and bacteria [12,13]. Over the last 10 years, Lake Baikal, the largest freshwater lake on Earth, located 10° to the south from LL and LV in latitudinal direction, has been actively studied with respect to microeukaryotic community features by applying a metagenomics approach and HTS techniques. Particularly, the under-ice phytoplankton communities of Lake Baikal were characterized using microscopy and HTS of SSU amplicons [14–18]. These results provide a valuable background that could be used to explore the microeukaryotic communities of LL/LV during the ice-covered and open water periods.

In this work, we compiled an initial inventory of unicellular eukaryotes in LL and LV during the ice cover period by using the metabarcoding of 18S rRNA and investigated the compositional differences between the sub-ice layer and the water column communities in these lakes. Furthermore, we evaluated the impact of seasonal changes on the community structure. Below, we discuss the consistency of the obtained results and hypothesize on the environmental factors influencing changes in community structure.

## 2. Materials and Methods

### 2.1. Sampling and Environmental Parameters

Detailed descriptions of the geographical position, climatic characteristics, physical and chemical environmental parameters of the Lake Vorota and Lake Labynkyr sampling sites were presented by Zakharova et al. [10]. Water samples were collected from the sub-ice layer of the bottom ice surface (SI) and from the water column (WC) during April and May 2016 and June 2017. For WC, two liters of water were sampled from 5 m depth using a Niskin bottle. The SI water samples were collected manually from the bottom ice surface (water layer thickness ~1–2 cm) by divers holding the syringe in the outstretched arm.

### 2.2. Amplicon Library Preparation, Sequencing, Raw Data Processing and Quality Control

Approximately 2 L out of each of 20 water samples (Table 1) was first prefiltered through a 27 μm mesh and then filtered through 0.2 μm polycarbonate membrane (Whatman Part of GE HealthCare, Chicago, Illinois, USA). Biomass collected on 0.2 μm membrane was washed out of the filters into sterile flasks filled with 5 mL of TE buffer (10 mM Tris-HCl, 1 mM EDTA; pH 7.5) and stored at −20 °C until samples were transported to the laboratory. Then, samples were stored at −80 °C until further analysis. DNA was extracted using lysozyme (1 mg/mL), proteinase K, 10% SDS, and phenol:chloroform:isoamyl alcohol mixture (25:24:1) according to the protocol based on Rusch et al. [19].

**Table 1.** Hydrophysical and hydrochemical characteristics of samples.

| Month | Sample | Temp | pH | EC | DO | $PO_4^{3-}$ | $NH_4^+$ | $NO_2^-$ | $NO_3^-$ | $N_{min}$ | TDS | TOC | TMA | TMB | Snow | Ice |
|-------|--------|------|-----|-------|------|-------|-------|-------|------|------|-------|------|--------|------|------|-----|
| April | L1SI04 | 0.4 | 8.28 | 49 | 10.6 | 0.001 | 0.021 | 0.002 | 0.35 | 0.1 | 33.19 | 3.26 | 19.57 | 0.07 | 30 | 91 |
| | L1WC04 | 1.2 | 7.84 | 43.22 | 10.6 | 0.003 | 0.059 | 0.002 | 0.3 | 0.12 | 29.29 | 2.58 | 13.7 | 0.03 | 30 | 91 |
| | L2SI04 | 1.2 | 7.71 | 45.98 | 7.5 | 0.001 | 0.062 | 0.002 | 0.34 | 0.13 | 31.82 | 2.88 | 24.1 | 0.06 | 32 | 86 |
| | L2WC04 | 2.6 | 7.93 | 39.49 | 7.5 | 0.002 | 0.058 | 0.001 | 0.29 | 0.11 | 26.67 | 2.21 | 35.95 | 0.01 | 32 | 86 |
| | L3SI04 | 0.4 | 7.55 | 44.92 | 8.9 | 0.001 | 0.014 | 0.002 | 0.33 | 0.08 | 30.72 | 3.45 | 20.95 | 0.02 | 30 | 109 |
| | L3WC04 | 1.9 | 7.93 | 39.49 | 8.9 | 0.002 | 0.058 | 0.001 | 0.29 | 0.11 | 26.67 | 2.21 | 28.85 | 0.03 | 30 | 109 |
| | L4SI04 | 0.4 | 7.48 | 50.1 | 8.2 | 0.004 | 0.019 | 0.001 | 0.42 | 0.11 | 33.96 | 3.04 | 40.4 | 0.04 | 35 | 86 |
| | L4WC04 | 3.7 | 7.7 | 39.85 | 8.2 | 0.001 | 0.069 | 0.002 | 0.28 | 0.12 | 26.54 | 3.04 | 39.55 | 0.13 | 35 | 86 |
| May | L1SI05 | 0.5 | 8.78 | 38.41 | 8.5 | 0.009 | 0.013 | 0.002 | 0.33 | 0.08 | 25.33 | 1.45 | 15.75 | 0.01 | 5 | 111 |
| | L1WC05 | 2.6 | 9.34 | 45.72 | 8.5 | 0.003 | 0.012 | 0.002 | 0.34 | 0.09 | 30.3 | 2.14 | 222.23 | 0.08 | 5 | 111 |
| | L3SI05 | 1.2 | 7.81 | 23.34 | 9.8 | 0.006 | 0.007 | 0.003 | 0.19 | 0.05 | 17.16 | 1.99 | 34.4 | 0.03 | 0.5 | 110 |
| | L3WC05 | 3.4 | 9.26 | 40.76 | 9.8 | 0.002 | 0.016 | 0.001 | 0.2 | 0.09 | 27.71 | 2.66 | 313.67 | 0.03 | 0.5 | 110 |
| | L4SI05 | 1.3 | 6.97 | 8.517 | 9.6 | 0 | 0.009 | 0.002 | 0.17 | 0.05 | 6.56 | 1.5 | 52.56 | 0.06 | 0.5 | 86 |
| | V1SI05 | 0.4 | 8.94 | 41.48 | 8.5 | 0.008 | 0.001 | 0.003 | 0.1 | 0.02 | 31.74 | 1.24 | 192.7 | 0.03 | 1 | 100 |
| | V1WC05 | 3.1 | 8.82 | 51.21 | 8.5 | 0.012 | 0.004 | 0 | 0.08 | 0.02 | 42.47 | 0.86 | 79.55 | 0.08 | 1 | 100 |
| June | L1WC06 | 3.6 | 6.8 | 30.89 | 8.3 | 0.003 | 0.017 | 0.003 | 0.33 | 0.09 | - | 1.05 | 173.3 | 0.02 | 5 | 110 |
| | L3WC06 | 2.5 | 6.97 | 37.71 | 7.3 | 0.003 | 0.012 | 0.003 | 0.38 | 0.09 | - | 0.86 | 127.1 | 0.03 | 1 | 80 |
| | L4WC06 | 5.6 | 6.98 | 41.07 | 8.4 | 0.016 | 0.012 | 0.005 | 0.45 | 0.11 | - | 0.86 | 121.3 | 0.05 | 0.5 | 86 |
| | V1SI06 | 0.4 | 7.15 | 49.55 | 6.7 | 0.014 | 0.015 | 0.003 | 0.05 | 0.02 | - | 0.83 | 55.67 | 0.02 | 5 | 110 |
| | V1WC06 | 3.2 | 7.21 | 54.75 | 6.7 | 0.023 | 0.016 | 0.006 | 0.06 | 0.02 | - | 1.84 | 95.13 | 0.04 | 5 | 110 |
| | V2WC06 | 2.2 | 7.1 | 33.8 | 8.3 | 0.014 | 0.016 | 0.003 | 0.04 | 0.03 | - | 0.75 | 84.91 | 0.02 | 5 | 83 |

Column legend: Month—month of sampling. Sample—Sample id: the sampling site is encoded by the first two symbols, with the first letter denoting Labynkyr or Vorota and the digit denoting the sampling point; the fourth and fifth letters denote the sampling layer: sub-ice (SI) or water column (WC); month is encoded by the last two digits. Temp—water temperature, °C; EC—water electric conductivity, $\mu S/cm^2$; DO—dissolved oxygen, mg/L; $PO_4^{3-}$—concentration of phosphate anion, mg/L; $NH_4^+$—concentration of ammonium cation, mg/L; $NO_2^-$—concentration of nitrite anion, mg/L; $NO_3^-$—concentration of nitrate anion, mg/L; $N_{min}^-$—total concentration of mineral nitrogen from ammonium, nitrate and nitrite, mg/L; TDS—total dissolved solids, mg/L; TOC—total organic carbon, mg/L; TMA—total microalgae abundance, $\times 10^3$ cells/L; TMB—total microalgae biomass, $g/m^3$; Snow—snow cover thickness at the moment of sampling, cm; Ice—ice thickness at the moment of sampling, cm.

The purified genomic DNA was sent to HTS provider for amplicon library preparation and sequencing. Amplicon libraries were prepared with primers V8f (5′-ATAACAGG-TCTGTGATGCCCT-3′) and 1510R (5′-CCTTCYGCAGGTTCACCTAC-3′) [20], targeting a broad set of eukaryotic taxa. Amplicon library preparation was performed as described in "Illumina 16S Metagenomic Library Preparation Guide #15044223 Rev. B". Libraries were analyzed by the Core Centrum "Genomic Technologies, Proteomics and Cell Biology", ARRIAM (Saint-Petersburg, Russia) using Illumina MiSeq Sequencer with Reagent Kit v3 to obtain 300 bp paired-end reads. DNA samples L4SI05 and L4WC06 were used to generate technical replicates L4SI05-02 and L4WC06-02 with independent amplicon library preparation and sequencing procedures (Table 2).

**Table 2.** Summary NGS-data statistics and alpha-diversity indices of LL and LV microeukaryotic communities.

| Sample | Reads | Richness | ACE | Shannon | Simpson | Inverse Simpson |
|---|---|---|---|---|---|---|
| L1SI04 | 14,781 | 155 | 168.37 | 3.18 | 0.90 | 10.11 |
| L1WC04 | 51,984 | 98 | 98.00 | 2.54 | 0.80 | 5.05 |
| L2SI04 | 10,123 | 99 | 102.69 | 3.21 | 0.92 | 11.95 |
| L2WC04 | 45,914 | 73 | 73.00 | 2.74 | 0.85 | 6.67 |
| L3SI04 | 12,274 | 124 | 133.40 | 2.69 | 0.86 | 7.34 |
| L3WC04 | 11,225 | 113 | 114.46 | 3.05 | 0.89 | 9.26 |
| L4SI04 | 10,601 | 123 | 124.47 | 3.10 | 0.90 | 9.53 |
| L4WC04 | 10,258 | 93 | 93.70 | 3.30 | 0.94 | 15.55 |
| L1SI05 | 20,979 | 75 | 77.40 | 0.99 | 0.31 | 1.45 |
| L1WC05 | 26,057 | 191 | 191.91 | 3.64 | 0.93 | 14.74 |
| L3SI05 | 11,460 | 106 | 114.76 | 1.62 | 0.54 | 2.19 |
| L3WC05 | 14,548 | 206 | 212.88 | 3.48 | 0.91 | 11.04 |
| L4SI05 * | 35,709 | 161 | 177.54 | 2.25 | 0.81 | 5.15 |
| L4SI05-2 * | 21,734 | 96 | 98.73 | 2.23 | 0.82 | 5.41 |
| V1SI05 | 17,205 | 244 | 257.79 | 2.90 | 0.82 | 5.64 |
| V1WC05 | 531 | - | - | - | - | - |
| L1WC06 | 17,685 | 213 | 224.49 | 3.31 | 0.90 | 9.74 |
| L3WC06 | 27,683 | 337 | 340.35 | 4.16 | 0.95 | 21.76 |
| L4WC06 * | 22,401 | 326 | 331.59 | 4.05 | 0.94 | 17.57 |
| L4WC06-2 * | 24,580 | 153 | 171.89 | 2.42 | 0.81 | 5.35 |
| V1SI06 | 7032 | 112 | 134.98 | 2.27 | 0.75 | 3.99 |
| V1WC06 | 14,788 | 133 | 134.42 | 2.91 | 0.82 | 5.58 |
| V2WC06 | 15,535 | 131 | 135.60 | 3.09 | 0.92 | 11.78 |

Column legend: Sample—sample id; Reads—number of reads after quality control; Richness—observed number of $OTUs_{0.03}$; ACE—Abundance-based coverage estimator index; Shannon—Shannon index; Simpson—Simpson index; Inverse Simpson—inverse Simpson index; *—L4SI05/L4SI05-2 and L4WC06/L4WC06-2 samples are technical replicates.

Analysis of sequencing data was performed in Userch v.10 (https://www.drive5.com/usearch/ (accessed on 25 January 2023)) [21], Vsearch v.2.9.1 (https://github.com/torognes/vsearch (accessed on 25 January 2023)) [22], and mothur v.1.43.10 (https://mothur.org/ (accessed on 25 January 2023)) [23]. Raw reads were merged, primer sequences were truncated, and contigs were filtered by the expected error threshold 1.0 with usearch (option -fastq_maxee 1.0). Next, sequences were clustered with vsearch command "-cluster_size" at 0.97 identity threshold, and OTUs with less than two reads were discarded. OTU centroids

were subjected to chimera filtering by UCHIME-denovo followed by UCHIME-reference of vsearch. Finally, the community composition matrix was generated by remapping of merged and quality-filtered reads to a chimera-free set of OTUs with identity threshold 0.97 (vsearch command "-usearch_global").

### 2.3. Bioinformatics, Statistical Analyses, and Data Visualization

OTU sequences were taxonomically classified using Silva v.138 (Bremen, Germany) [24] in mothur with probability cutoff set to 80 and further filtered by taxonomy to drop chloroplast-specific, mitochondria-specific and multicellular eukaryotic OTUs as well as OTUs with Kingdom = "unknown". Additionally, the taxonomy mapping was performed with the online ACT service https://www.arb-silva.de/aligner (25 February 2023). Results of taxonomic assignment and sequence similarity search are listed in Table S3. To generate the summary statistics on the abundance of phylotypes (Table S2), the raw community composition matrix was standardized by a median of sample sequencing depth, OTUs with total abundance less than 10 were filtered out as low-covered, phylotype counts were aggregated up to the "Class" taxonomic level, which was manually assigned using the results of mothur/ACT taxonomic pipelines (Table S4), and the top-13 abundant classes were selected to draw the histogram.

Statistical analyses were performed in R. The rarefaction curves, ACE index (non-parametric species richness estimators), and Shannon, Simpson, and inverse Simpson indices were calculated to measure the alpha diversity. OTUs with a total abundance below 10 were excluded from the community composition matrix prior to these calculations. Exploratory analyses of community composition were performed using vegan v.2.5-6 [25], phyloseq [26], and pvclust [27] (https://www.R-project.org/ (accessed on 25 January 2023)).

One-way ANOVA and Kruskal–Wallis statistical tests were used to examine the impact of independent factor variables on the discrete/continuous environmental variables (Figure S1) and alpha-diversity metrics (Figure S2) associated with the microeukaryotic community profiles. Two-way ANOVA test was used to estimate the main effect of factors "Month" and "Layer" and the effect of their interaction. We calculated two-way ANOVA statistics with R function "Anova" from package "car", using the "type-III" test for experiments with unbalanced design. The *p*-values, which were obtained by testing the independent factor variable (or two of them) versus a set of dependent variables (14 environmental variables and five alpha-diversity metrics), were adjusted using the false discovery rate (FDR) procedure. Both raw and adjusted *p*-values were reported. May and June groups were merged into "MJ" category when the alpha-diversity metrics were tested by ANOVA and Kruskal–Wallis procedures for the following reasons: (i) the SI June group was presented by a single profile with values falling into the range of SI May profiles; (ii) WC May group was presented by two profiles with values within the range of WC June profiles (Figure S2(C1–D5)). R-package ggpubr v. 0.6.0 (https://cran.r-project.org/web/packages/ggpubr/index.html (accessed on 25 January 2023)) was used to visualize the group-wise distribution of dependent variables.

For exploratory analyses, OTUs with a relative abundance above 0.1% or standardized abundance above 50 were selected to produce a count matrix. OTU counts were transformed with $\log(x + 1)$ and subjected to the transformation-based principal coordinate analysis (tb-PCoA). Linear regression of explanatory variables was performed by the envfit function of package vegan, followed by adjustment of permutation-based regression *p*-values by the Holm procedure. Environmental factors having an adjusted *p*-value threshold below 0.05 were drawn on the ordination plane. The pairwise distance matrix computed with Bray–Curtis similarity index was used to cluster communities by UPGMA. A robustness of clustering dendrogram was accessed by bootstrap- and au-values in the pvclust package.

The transformation-based redundancy analysis (tb-RDA) was used to evaluate the impact of environmental factors on species composition. OTUs with relative abundance above 0.1% were selected to produce a count matrix. OTU counts were transformed with

log(x + 1) and subjected to tb-RDA. Explanatory variables were chosen by a "forward selection" approach, followed by further filtering on *p*-value and the impact of the model ability to explain the total variance (Table S1).

DESeq2 package [28] was used to evaluate the significance of OTU abundance differences between the groups of samples. The raw community composition matrix of the non-transformed OTU counts, which was generated by the quality control/clustering pipeline described above (usearch-vsearch-taxonomic filtering), was used as DESeq2 input. The grouping of samples for differential abundance testing was established considering the results of exploratory analyses. It included five subsets of microeukaryotic community profiles: (i) LL SI April; (ii) LL SI May; (iii) LL WC April; (iv) LL WC May–June; (v) LV SI/WC June (all LV profiles except V1SI05). Ten contrasts corresponding to pairwise combinations of these five subsets were used for OTU differential abundance analysis. A phyloseq object was transformed to DESeq2 object, followed by estimation of size factors with geometric means of OTU counts, "local" dispersion estimate, and computation of Wald test statistics. The FDR-adjusted *p*-value and log-fold change (LFC) thresholds were set to 0.05 and $\log_2(1.5)$, respectively. OTUs indicating significant difference between groups of samples are listed in Table S5.

## 3. Results

### 3.1. Environmental Parameters at Sampling Sites

The lake ice was covered by snow at the time of sampling, with its thickness decreasing from 30–35 cm in April to 0.5–5 cm in June. Samples from the LL ice bottom surface were not collected in June, as the process of active ice degradation started by this time, and vertical channels filled with water were formed in the ice. During the sampling period, the water temperature of the SI layer varied between 0.4 and 1.2 °C, while the WC temperature range was 1.2–5.6 °C. The pH value varied between 6.8 and 9.34, electric conductivity was between 8.5 and 55 $\mu$S/cm$^2$, and mineral nitrogen concentration ranged between 0.02 and 0.13 mg/L (Table 1).

### 3.2. Quality Control of the Amplicon Clustering Results and Rarefaction Analysis

Sample V1WC05 was excluded from further analysis due to the lack of coverage for targeted microeukaryotic taxa: almost all OTUs of V1WC05 were classified as Arthropoda-specific sequences. For the remaining 20 samples, 22 amplicon data bunches were generated, including pairs of technical replicates for L4WC06 and L4SI05 (Table 2). After quality control steps, the coverage per sample varied between 7032 and 51,984 microeukaryote-specific reads. Processing of the amplicon sequencing data produced 459 distinct OTUs (3% dissimilarity threshold, minimal OTU coverage above 10 reads), ranging from 73 to 337 OTUs per sample with mean sequence length of 330 bp. Rarefaction curves of the amplicons reached saturation, indicating the diversity of the microeukaryotic V8-V9 SSU fragments was decently recovered, at least under current experimental settings (Figure 1). The exception was V1SI06, which had minimal sample coverage for microeukaryote-specific taxa because of contamination by Arthropoda-specific sequences.

### 3.3. Factors Influencing the Hydrophysical and, Hydrochemical Characteristics of Samples and the Alpha-Diversity Metrics of Microeukaryotic Communities

ANOVA revealed the phosphate and nitrate anion concentrations, and $N_{min}$ were significantly different between LL and LV samples (Table 3, Figure S1(A5,A8,A9)). The phosphate anion concentration was higher in LV, while the nitrate anion and $N_{min}$ concentrations were higher in LL. The concentrations of TOC and nitrite anion by LL/LV showed a barely significant response (Figure S1(A7,A10)). A number of hydrochemical, hydrophysical and environmental parameters, specifically pH, phosphate, ammonium, nitrite, $N_{min}$, TOC, TMA and the snow thickness, varied from April to June (Table 3, Figure S1(B1–B14)). The phosphate and nitrite anion concentrations, as well as TMA, gradually increased from April to June (Figure S1(B5,B7,B11)). The ammonium cation, $N_{min}$ and TOC concentrations

decreased from April to June (Figure S1(B6,B9,B10)). However, the only variable differing between layers was water temperature (Table 3, Figure S1(C1)), which obviously was lower in the SI layer than in the WC layer.

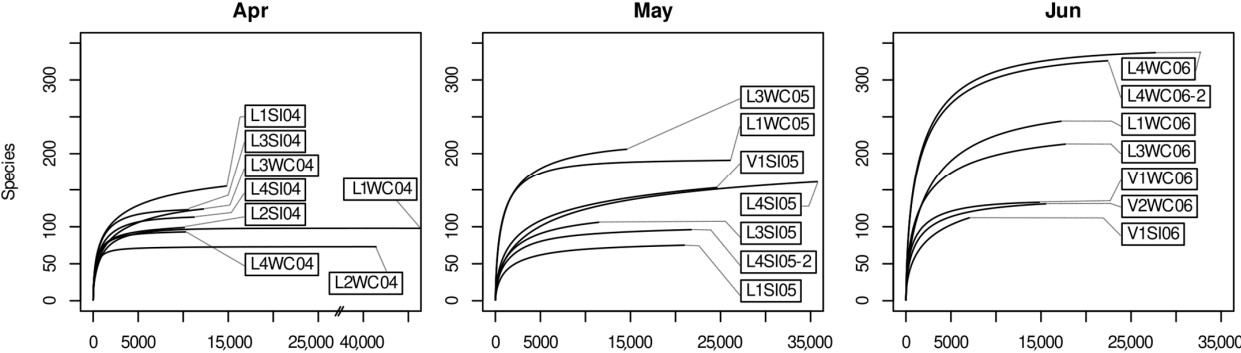

**Figure 1.** Rarefaction curves of microeukaryotic communities. Curves are grouped by the month of sampling.

Importantly, the two-factor sorting of environmental variables by the sampling layer and month also produced a set of distinct spatiotemporal patterns. In the WC layer, the pH, concentration of ammonium cation, concentration of nitrite anion, and TMA differed significantly over time (Table 3, Figure S1(D2,D6,D7,D11)), reinforcing the simple temporal response (Figure S1(B2,B6,B7,B11). Similarly, in SI layer, the concentrations of nitrate anion and $N_{min}$ decreased over time (Figure S1(D8,D9)). The layer-wise temporal concentrations of TOC basically mirrored its general temporal dynamics (Table 3, Figure S1(B10,D10)), with some lag in the WC layer.

The alpha-diversity metrics were also detected to be different over the sampling layer and month factors: the Shannon, Simpson and inverse Simpson indices were higher in WC profiles (Table 3, Figure S2(C3,C5)), while the species richness and ACE were gradually increased from April to June (Table 3, Figure S2(B1,B2)). The two-factor sorting of alpha-diversity metrics by the sampling layer and month also produced discrete patterns: the species richness and ACE were significantly increased in May–June WC profiles, while the Shannon and inverse Simpson indices were higher in April SI profiles (Table 3, Figure S2(D1,D5)).

With respect to interaction of the sampling layer and month factor variables, two-way ANOVA produced barely significant results for two out of 14 environmental variables: concentration of TOC and TMA (Table 4). At the same time, Shannon and Simpson alpha-diversity metrics displayed good support for the two-factor interaction model, which included the sampling layer and month factors. Additionally, the species richness, ACE, and the inverse Simpson metrics revealed that the "Month:Layer" interaction term of the two-factor model was barely significant (Table 4).

*3.4. Overall Community Composition and Comparison of Microeukaryotic Profiles*

The generated OTUs were taxonomically assigned to 34 classes of microeukaryotes. This set represented a wide spectrum of unicellular protists. The most abundant were (in decreasing order) the classes Dinophyceae, Chrysophyceae, Ciliophora, Trebouxiophyceae (green algae), Bacillariophyceae, Basidio- and Ascomycota (Fungi), unclassified Chlorophyta, Cercozoa, Cryptophyceae, Prymnesiophyceae (Haptophyta), and Telonena (SAR group) (Figure 2). Ciliophora were abundant in April LL profiles, with decreasing counts in June. Diatoms were predominantly abundant in WC communities of May–June. A distinctive feature of LV profiles was the increased number of Trebouxiophyceae, which were also present in June LL WC, while their abundance was lower. Peaks of Basidiomycota and Cercozoa in LL occurred in April, with Basidiomycota predominantly abundant in the WC layer. On the other hand, Ascomycota predominantly proliferated in May communities.

**Table 3.** One-way ANOVA computed for environmental variables and alpha-diversity metrics.

| Indep * Var #1 | Indep Var #2 | Dep Var | ANOVA | | | Kruskal–Wallis | | |
|---|---|---|---|---|---|---|---|---|
| | | | $p$ | $p_{adj}$ | $p_{adj}$ Sign | $p$ | $p_{adj}$ | $p_{adj}$ Sign |
| | | | *Environmental variables* | | | | | |
| Lake | - | $PO_4$ | $2.64 \times 10^{-4}$ | $1.23 \times 10^{-3}$ | ** | $7.56 \times 10^{-3}$ | $3.53 \times 10^{-2}$ | * |
| Lake | - | $NO_2$ | $1.81 \times 10^{-2}$ | $6.33 \times 10^{-2}$ | . | $1.48 \times 10^{-2}$ | $5.19 \times 10^{-2}$ | . |
| Lake | - | $NO_3$ | $6.52 \times 10^{-6}$ | $5.49 \times 10^{-5}$ | *** | $2.44 \times 10^{-3}$ | $1.71 \times 10^{-2}$ | * |
| Lake | - | totN | $7.84 \times 10^{-6}$ | $5.49 \times 10^{-5}$ | *** | $2.25 \times 10^{-3}$ | $1.71 \times 10^{-2}$ | * |
| Lake | - | TOC | $3.39 \times 10^{-2}$ | $9.49 \times 10^{-2}$ | . | $2.32 \times 10^{-2}$ | $6.49 \times 10^{-2}$ | . |
| Month | - | pH | $7.36 \times 10^{-4}$ | $2.49 \times 10^{-3}$ | ** | $5.73 \times 10^{-3}$ | $1.61 \times 10^{-2}$ | * |
| Month | - | DO | $3.76 \times 10^{-2}$ | $5.84 \times 10^{-2}$ | . | $2.71 \times 10^{-2}$ | $4.21 \times 10^{-2}$ | * |
| Month | - | $PO_4$ | $2.92 \times 10^{-3}$ | $6.81 \times 10^{-3}$ | ** | $1.22 \times 10^{-2}$ | $2.13 \times 10^{-2}$ | * |
| Month | - | $NH_4$ | $6.06 \times 10^{-4}$ | $2.49 \times 10^{-3}$ | ** | $1.86 \times 10^{-3}$ | $7.58 \times 10^{-3}$ | ** |
| Month | - | $NO_2$ | $8.88 \times 10^{-4}$ | $2.49 \times 10^{-3}$ | ** | $2.16 \times 10^{-3}$ | $7.58 \times 10^{-3}$ | ** |
| Month | - | Nmin | $6.54 \times 10^{-3}$ | $1.31 \times 10^{-2}$ | * | $7.66 \times 10^{-3}$ | $1.79 \times 10^{-2}$ | * |
| Month | - | TOC | $7.00 \times 10^{-6}$ | $4.90 \times 10^{-5}$ | *** | $6.49 \times 10^{-4}$ | $4.54 \times 10^{-3}$ | ** |
| Month | - | TMA | $2.20 \times 10^{-2}$ | $3.86 \times 10^{-2}$ | * | $1.21 \times 10^{-2}$ | $2.13 \times 10^{-2}$ | * |
| Month | - | Snow | $4.15 \times 10^{-15}$ | $5.82 \times 10^{-14}$ | *** | $6.03 \times 10^{-4}$ | $4.54 \times 10^{-3}$ | ** |
| Layer | - | Temp | $2.81 \times 10^{-5}$ | $3.93 \times 10^{-4}$ | *** | $2.69 \times 10^{-4}$ | $3.77 \times 10^{-3}$ | ** |
| WC | Month | pH | $8.98 \times 10^{-8}$ | $2.52 \times 10^{-6}$ | *** | $1.34 \times 10^{-2}$ | $9.87 \times 10^{-2}$ | . |
| SI | Month | $PO_4$ | $2.89 \times 10^{-2}$ | $7.35 \times 10^{-2}$ | . | $1.66 \times 10^{-1}$ | $2.74 \times 10^{-1}$ | . |
| WC | Month | $NH_4$ | $1.87 \times 10^{-7}$ | $2.53 \times 10^{-6}$ | *** | $2.52 \times 10^{-2}$ | $9.87 \times 10^{-2}$ | . |
| WC | Month | $NO_2$ | $1.64 \times 10^{-2}$ | $4.72 \times 10^{-2}$ | * | $1.89 \times 10^{-2}$ | $9.87 \times 10^{-2}$ | . |
| SI | Month | $NO_3$ | $1.54 \times 10^{-2}$ | $4.72 \times 10^{-2}$ | * | $4.50 \times 10^{-2}$ | $9.87 \times 10^{-2}$ | . |
| SI | Month | Nmin | $1.69 \times 10^{-2}$ | $4.72 \times 10^{-2}$ | * | $5.00 \times 10^{-2}$ | $1.00 \times 10^{-1}$ | . |
| SI | Month | TOC | $2.52 \times 10^{-4}$ | $1.17 \times 10^{-3}$ | ** | $3.57 \times 10^{-2}$ | $9.87 \times 10^{-2}$ | . |
| WC | Month | TOC | $1.76 \times 10^{-3}$ | $7.06 \times 10^{-3}$ | ** | $2.20 \times 10^{-2}$ | $9.87 \times 10^{-2}$ | . |
| WC | Month | TMA | $1.41 \times 10^{-4}$ | $7.90 \times 10^{-4}$ | *** | $1.36 \times 10^{-2}$ | $9.87 \times 10^{-2}$ | . |
| SI | Month | Snow | $3.99 \times 10^{-6}$ | $2.80 \times 10^{-5}$ | *** | $3.80 \times 10^{-2}$ | $9.87 \times 10^{-2}$ | . |
| WC | Month | Snow | $2.72 \times 10^{-7}$ | $2.53 \times 10^{-6}$ | *** | $2.39 \times 10^{-2}$ | $9.87 \times 10^{-2}$ | . |
| | | | *Alpha-diversity metrics* ** | | | | | |
| Month | - | Richness | $3.95 \times 10^{-2}$ | $9.87 \times 10^{-2}$ | . | $1.54 \times 10^{-2}$ | $3.84 \times 10^{-2}$ | * |
| Month | - | ACE | $2.98 \times 10^{-2}$ | $9.87 \times 10^{-2}$ | . | $1.16 \times 10^{-2}$ | $3.84 \times 10^{-2}$ | * |
| Layer | - | Shannon | $3.16 \times 10^{-3}$ | $1.58 \times 10^{-2}$ | * | $6.86 \times 10^{-3}$ | $3.43 \times 10^{-2}$ | * |
| Layer | - | Simpson | $3.74 \times 10^{-2}$ | $6.24 \times 10^{-2}$ | . | $2.50 \times 10^{-2}$ | $4.16 \times 10^{-2}$ | * |
| Layer | - | Inverse Simpson | $2.00 \times 10^{-2}$ | $4.99 \times 10^{-2}$ | * | $2.50 \times 10^{-2}$ | $4.16 \times 10^{-2}$ | * |
| WC | Month | Richness | $2.71 \times 10^{-2}$ | $6.78 \times 10^{-2}$ | . | $6.58 \times 10^{-3}$ | $2.10 \times 10^{-2}$ | * |
| WC | Month | ACE | $2.30 \times 10^{-2}$ | $6.78 \times 10^{-2}$ | . | $6.58 \times 10^{-3}$ | $2.10 \times 10^{-2}$ | * |
| SI | Month | Shannon | $6.84 \times 10^{-3}$ | $3.42 \times 10^{-2}$ | * | $1.05 \times 10^{-2}$ | $2.10 \times 10^{-2}$ | * |
| SI | Month | Inverse Simpson | $1.03 \times 10^{-3}$ | $1.03 \times 10^{-2}$ | * | $1.05 \times 10^{-2}$ | $2.10 \times 10^{-2}$ | * |

Column legend: Indep Var #1/2—independent variables used to divide the dataset into subgroups; Dep Var—tested dependent variable; ANOVA/Kruskal–Wallis *p*-values; *p*—*p*-value; $p_{adj}$—FDR-adjusted *p*-value; $p_{adj}$ sign—the adjusted *p*-value significance code: *** ≤ 0.001; 0.001 ≤ ** ≤ 0.01; 0.01 ≤ * ≤ 0.05; 0.05 ≤ . ≤ 0.1; *—only rows with ANOVA $p_{adj} \leq 0.1$ are shown, i.e., cases where the factor-wise distributions of dependent variable differ significantly; **—May and June groups were merged when the alpha-diversity metrics were tested by ANOVA.

**Table 4.** Two-way ANOVA with interaction of factors computed for environmental variables and alpha-diversity metrics.

| Factor * | Dep Var | $p$ | $p_{adj}$ | $p_{adj}$ Sign |
|---|---|---|---|---|
| *Environmental variables* | | | | |
| Layer | Temp | $1.83 \times 10^{-2}$ | $9.61 \times 10^{-2}$ | . |
| Month | EC | $1.47 \times 10^{-2}$ | $8.84 \times 10^{-2}$ | . |
| Layer | NH4 | $9.94 \times 10^{-4}$ | $1.39 \times 10^{-2}$ | * |
| Month | Nmin | $1.01 \times 10^{-2}$ | $7.05 \times 10^{-2}$ | . |
| Month | TOC | $1.53 \times 10^{-5}$ | $3.22 \times 10^{-4}$ | *** |
| Month:Layer | TOC | $8.00 \times 10^{-3}$ | $6.72 \times 10^{-2}$ | . |
| Month:Layer | TMA | $7.64 \times 10^{-3}$ | $6.72 \times 10^{-2}$ | . |
| Month | Snow | $1.58 \times 10^{-10}$ | $6.63 \times 10^{-9}$ | *** |
| *Alpha-diversity metrics ** * | | | | |
| Month:Layer | Richness | $3.61 \times 10^{-2}$ | $8.30 \times 10^{-2}$ | . |
| Month:Layer | ACE | $3.87 \times 10^{-2}$ | $8.30 \times 10^{-2}$ | . |
| Month | Shannon | $2.16 \times 10^{-3}$ | $1.62 \times 10^{-2}$ | * |
| Month:Layer | Shannon | $8.96 \times 10^{-4}$ | $1.34 \times 10^{-2}$ | * |
| Month | Simpson | $8.41 \times 10^{-3}$ | $4.20 \times 10^{-2}$ | * |
| Month:Layer | Simpson | $2.57 \times 10^{-2}$ | $8.19 \times 10^{-2}$ | . |
| Month | Inverse Simpson | $4.54 \times 10^{-2}$ | $8.51 \times 10^{-2}$ | . |
| Month:Layer | Inverse Simpson | $2.73 \times 10^{-2}$ | $8.19 \times 10^{-2}$ | . |

Column legend: Factor (Month, Layer or Month:Layer)—independent variable or their interaction; Dep Var—tested dependent variable; $p$—$p$-value; padj—FDR-adjusted $p$-value; $p_{adj}$ sign—the adjusted $p$-value significance code: *** $\leq 0.001$; $0.001 \leq$ ** $\leq 0.01$; $0.01 \leq$ * $\leq 0.05$; $0.05 \leq$ . $\leq 0.1$; *—only rows with $p_{adj} \leq 0.1$ are shown, i.e., cases where the two-factor interaction model has the significant impact to the dependent variable; **—May and June groups were merged when the alpha-diversity metrics were tested by ANOVA.

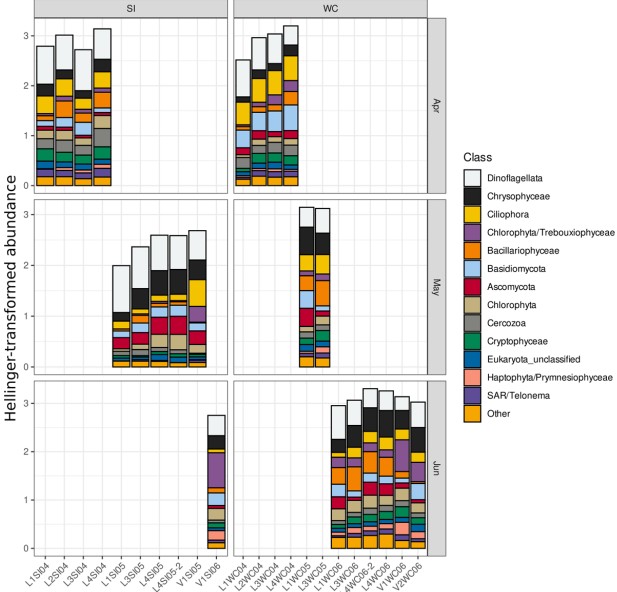

**Figure 2.** Hellinger-transformed relative abundances of the top-13 microeukaryotic classes in LL and LV communities. The bar stacks are arranged into six facets by two factor variables: "Layer" and "Month". The profiles are horizontally sorted by sampling site, month and sampling layer. Taxa presented in the color legend are sorted by total abundance in decreasing order, except the last category "Other". The order of Classes in each bar is the same as in the color legend.

Exploratory analyses revealed that the major factor variables influencing the profile clustering were the month of sampling (April/May–June) and the sampling layer (SI/WC) (Figure 3). In tb-PCoA, the LL SI April communities were similar to the LL WC April ones (Figure 3A,C, quadrant III). The LL SI May profiles were completely different from others, forming a distinct group (Figure 3A,C, quadrant II). The LL WC microeukaryotic communities of May–June clustered together (Figure 3A,C, quadrant IV).

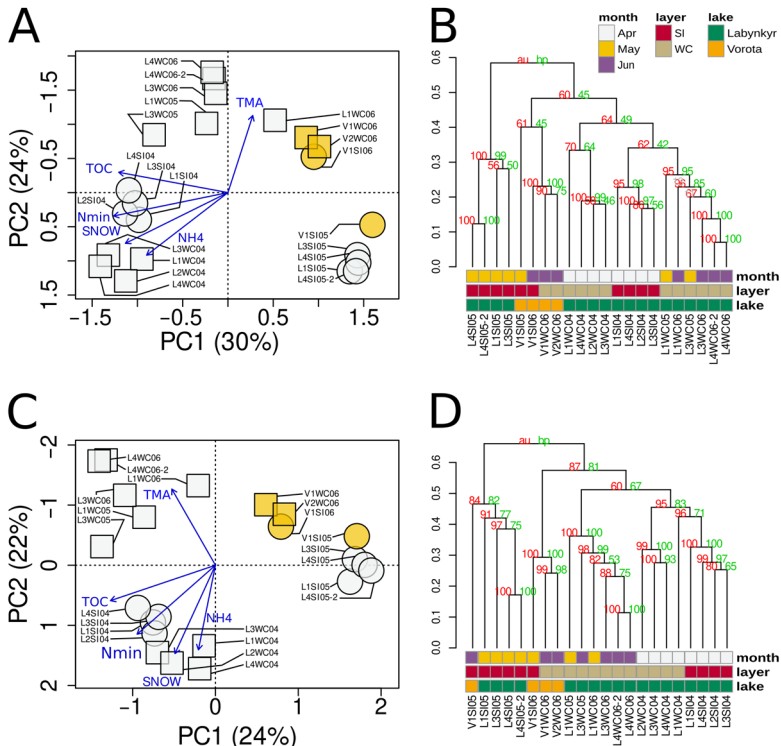

**Figure 3.** Exploratory analysis of LL and LV microeukaryotic community profiles. Upper and bottom rows of the image were generated using different minimum OTU abundance thresholds: OTU relative abundance $\geq$0.1% ("major OTUs") (**A**,**B**) and OTU standardized total abundance $\geq$50 (**C**,**D**). The former OTU count matrix contains 95 top-abundant OTUs with minimum OTU standardized total abundance $\geq$361. The latter count matrix consists of 253 OTUs. (**A**,**C**) tb-PCoA ordination biplots. Point shape: SI—circle, WC—square. Point color: LL—grey, LV—yellow. Blue arrows—linear regression of explanatory variables, showing the direction and range of their impact. (**B**,**D**) UPGMA-assisted community profile clustering dendrograms based on Bray–Curtis pairwise distance matrices. Red and green numbers indicate the "approximately unbiased" *p*-values and "bootstrap probability" values, respectively. Color annotation above the sample IDs encodes the spatial and temporal sampling parameters.

Analysis of alpha diversity (Tables 2 and 3) showed that the LL WC microeukaryotic communities of May–June had the highest richness. Noticeably, the diversity of the May–June WC microeukaryotic community profiles was also increased in comparison with the May–June SI profiles. Using the results of differential abundance analysis (Table S5) and OTU abundance heatmap (Figure 4), one may identify the key OTUs that are responsible for this effect. Particularly, OTU146 (family Suessiaceae), OTU153 (Chlorophyta), OTU305 (Chrysophyceae), OTU442 (order Perkinsidae), OTU115 (Chrysophyceae), and OTU576 (order Eustigmatales) were detected in the LL WC May–June profiles, while the rest of the communities lacked these OTUs. Typically, the LL WC May–June profiles were the most similar to those of LL SI April. The latter group was highly similar to LL WC April (Figure 3B,D). The common feature of these groups was the increased abundance of OTUs in the middle part of Figure 4 (top-down from OTU45 to OTU221). Species classified as

Perkinsidae, Chrysophyceae, Choanoflagellata, Chlorophyta, Intramacronucleata (orders Spirotrichea and Conthreep), Dinophyceae and Cercozoa were presented in this subset. At the same time, several OTUs from this subset (top-down from OTU56 to OTU221), which were classified as Dinophyceae, Chlorophyta, Chlorodendrophyceae, Intramacronucleata, Prymnesiophyceae, MAST-2, and MAST-12, had increased abundance in the LV June profiles.

OTUs that were present in almost all profiles at top-abundant positions were taxonomically assigned to dinoflagellates (OTU3, OTU50, order Peridiniophyceae—OTU71), diatoms (OTU1), Cryptophyceae (OTU17), Ciliophora (order Spirotrichea—OTU42), Chrysophyceae (OTU49), Cercozoa (OTU15), and various fungi: Cystobasidiomycetes (OTU7), Leotiomycetes (OTU163), Dothideomycetes (OTU97, OTU117), and Tremellomycetes (OTU47).

### 3.5. Differentially Abundant OTUs of LL Communities in April

A comparatively small subset of differentially abundant OTUs with the relative abundance above 0.1% was found between the April SI and WC microeukaryotic community profiles (Figure 4). Eight OTUs were abundant in SI, while three were abundant in WC (Table S5).

The following OTUs were abundant in the LL SI profiles: OTU3, OTU65, OTU12, OTU158, OTU13, OTU27, OTU42, and OTU109. OTU3 was one of the most abundant sequences in the dataset. This dinoflagellate-specific OTU sequence (Table 5), which was classified as Peridiniphycidae, was abundant in the SI April and May profiles, as well as in the WC May–June communities. Dinoflagellate-specific OTU65, chrysophycean OTU12 and OTU158 had a high similarity with sequences found in freshwater ecosystems (Table 5).

The Ciliophora-specific OTUs from the SI April profiles were classified as free-living ciliates of classes Oligohymenophorea (OTU13), Oligotrichia (OTU27), uncultured ciliates (OTU42) and Haptotia (OTU109). These OTUs were similar to environmental sequences from freshwater ecosystems, including those from cold habitats (Table 5). The LL WC April profiles were abundant in OTU22, OTU48 and OTU97, possessing similarity with Ciliophora-, chrysophyte- and fungal-specific sequences (Table 5).

### 3.6. Differentially Abundant OTUs of LL Communities in May–June

The species diversity of LL SI communities was decreased in comparison with that of WC (Table 2, Figures 3A,B and 4). Out of 52 differentially abundant OTUs (relative abundance above 0.1%), 45 had increased abundance in the WC profiles, while seven OTUs were more abundant in SI (Table S5). A majority of OTUs, which were differentially abundant in the WC May profiles, possessed small absolute counts. Thus, the WC May profiles were characterized by increased species richness and diversity at the cost of appearance of a number of minor species. The exceptions from this pattern were the diatom-specific OTU1 and the chrysophyte-specific OTU21 (Table 5). Diatoms started massive development in late May in LL WC, while they were also abundant in the LL SI April profiles. Chrysophytes had their peak abundance in WC during May–June in both LL and LV, while they also were present in the SI May profiles.

Among the OTUs abundant in SI during May–June, there were sequences that were taxonomically assigned to dinoflagellates, chrysophytes, Cercosoa, ciliates and Fungi. The most abundant of these were Dinophyta-specific (OTU3 and OTU50) and chrysophyte-specific OTU404. As already noted, the LL SI April profiles had considerable similarity with the LL WC May–June ones. For a major number of microeukaryotic communities, similarity between the LL SI April and LL WC May–June groups was even more prominent, than that between the LL SI April and LL WC May (Figures 3B and 4) groups. One may argue for a scenario where the communities of microeukaryotes, which were formed in the SI layer in April, migrate into the water column once the ice starts to degrade. Thus, the SI April communities might represent the primer stage for development of the summer microeukaryotic communities in Labynkyr.

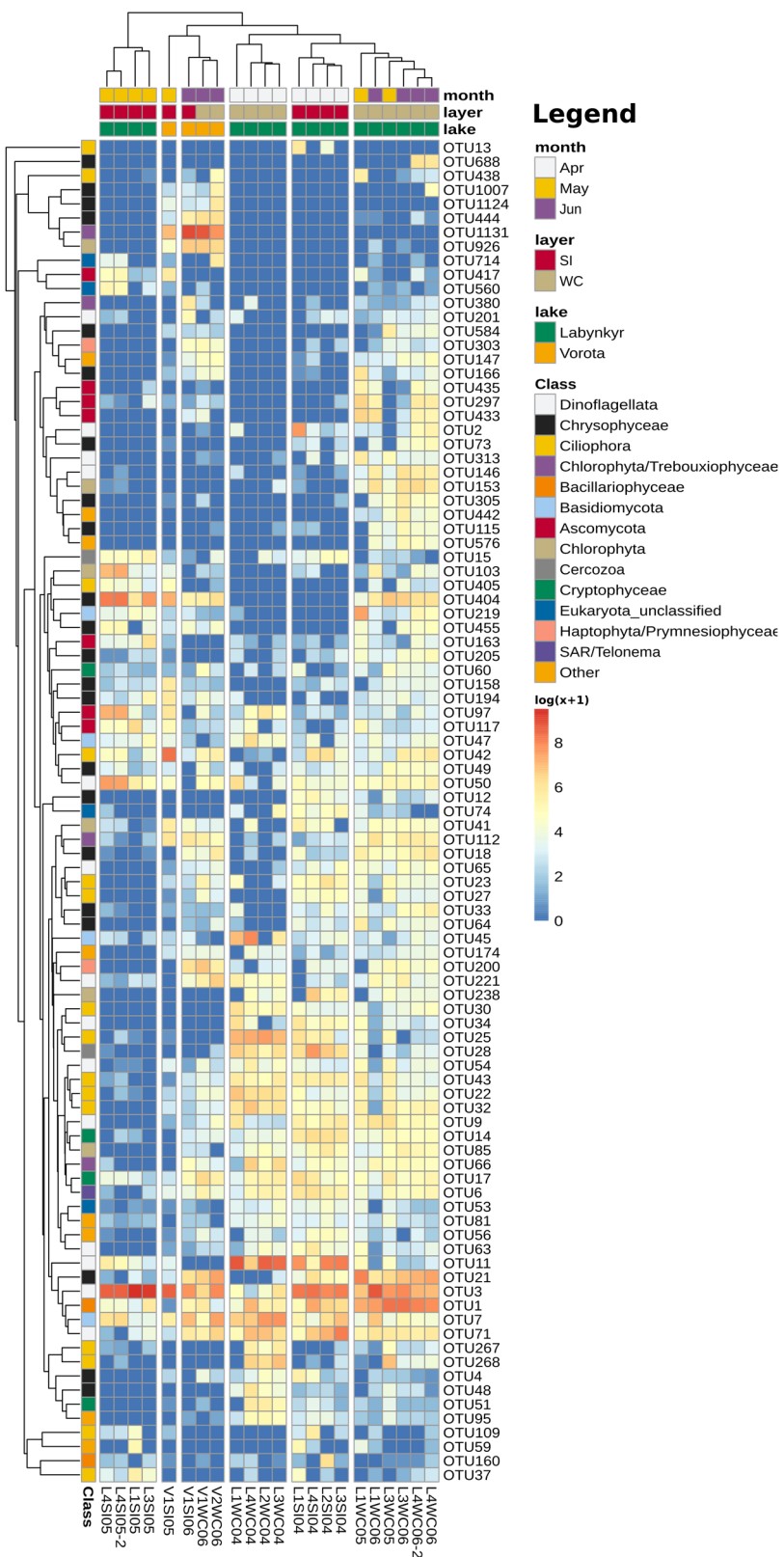

**Figure 4.** Heatmap of LL and LV microeukaryotic community profiles generated with a set of 95 top-abundant OTUs (see Figure 3A,B). Order of communities (columns) as in Figure 3B. UPGMA-assisted clustering of the Bray–Curtis pairwise distance matrices was used to produce the sample-wise and OTU-wise clustering trees. Color annotations above the heatmap describe the spatial and temporal categories of communities. Color annotation on the left denotes the OTU taxonomic affiliation on the "Class" level and is the same as in Figure 2.

**Table 5.** Environmental sources of selected SSU sequences having high-similarity matches with OTUs from LL and LV.

| OUT Number | Nearest Match | GenBank Accession | Sequence Identity, % | Source |
|---|---|---|---|---|
| OTU3 | *Scrippsiella hangoei* strain SHTV6 *Peridinium aciculiferum* strain PAER-2 | EF417316 EF417314 | 99 | The Gulf of Finland, Baltic Sea Lake Erken, Sweden [29] |
| OTU65 | Uncultured alveolate | DQ244019 | 99 | Lake Pavin, France [30] |
| OTU12 | *Chrysolepidomonas dendrolepidota* CCMP293Spumella-like isolate JBM19 | AF123297 FR865768 | 98 | Lake Superior, Keeweenaw Country, MI, USA [31] Lake Hallstatt, Austria [32] |
| OTU158 | Bacterivorous protozoa | AB749149 | 97 | Hirose River, Japan [33] |
| OTU13 | Free-living ciliate of class Oligohymenophorea | LR025746 HQ219368 | 98 | Lake Zurich, Switzerland Lake Aydat, France [34] |
| OTU27 | *Cyrtostrombidium longisomum* | KJ534582 | 97 | Coastal waters of northeastern Taiwan [35] |
| OTU42 | Uncultured ciliate Uncultured ciliate *Limnostrombidium viride* SW2012122001 | GU067975 LC165025 KU525754 | 99 | Esch-sur-Sure, Luxembourg Biwa, JapanZhanjiang, Guangdong province, China [36] |
| OTU109 | Uncultured eukaryote | AB695505 | 99 | Freshwater lake, East Antarctica [37] |
| OTU1 | *Stephanodiscus* sp. | AB430594 | 99 | Japan and South Korea [38] |
| OTU21 | Chrysophycean isolate | AY082970 | 94 | High-latitude Arctic lakes, Ellesmere Island [39] |
| OTU50 | *Woloszynskia pascheri* | EF058253 | 96 | Freshwater environment [40] |
| OTU444 | *Paraphysomonas foraminifera* strain TPC2 | AY651096 | 99 | Lake Mondsee, Austria [32] |
| OTU926 | *Ankyra lanceolata* strain Hg 1998-5 | AF302769 | 98 | Channel of Danube, Hungary [41] |
| OTU1131 | *Choricystis* sp. AS-29, green alga symbiont of sponges | AY195972 | 99 | Freshwater, free-living [42] Lake Baikal, sponge symbiont [43] |

### 3.7. Microeukaryotic Communities of LV and Their Comparison with LL Communities

There were four microeukaryotic community profiles in our dataset sampled from LV (Table 1). V1SI05 was collected in May 2015, while the three remaining samples (V1SI06, V1WC06 and V2WC06) were collected in June 2017. Although the LV communities were insufficiently sampled for in-depth statistical analysis in terms of their spatial and temporal variability, it seemed appropriate to use these results to compare the LV profiles between each other and with those of LL. The most prominent feature of the LV community profiles was the increased abundance of several OTUs, which were absent or very minor in the LL profiles (Figure 4, the upper part of heatmap). OTU1131 (Trebouxiophyceae), OTU444 (Chrysophyceae, genus *Paraphysomonas*), and OTU926 (Chlorophyceae) of this group were differentially abundant, as revealed by DESeq2 results (Table S5). These OTUs had sequences similar to those of chrysophytes and green algae from cold freshwater environments (Table 5).

### 3.8. Environmental Factors as Quantitative Explanatory Variables of Microeukaryotic Community Structure

We used the following continuous environmental variables to build a model of factors influencing the community structure: thickness of snow cover at sampling site; water temperature, pH, and electric conductivity (EC); concentrations of dissolved oxygen, $NH_4^+$, $PO_4^{3-}$, $NO_3^-$, $NO_2^-$ ions, total mineral nitrogen ($N_{min}$), total organic carbon (TOC); total microeukaryote abundance (TMA), and biomass (TMB) (Table 1). In tb-PCA, variability

explained by community composition was 67.5%. Searching for the most important environmental predictors with forward selection, the variability explained was 49.2%, and variables selected were the snow thickness, mineral nitrogen, electric conductivity, and water temperature (Table S1, Figure 5A). The ordination pattern obtained with the constrained approach was similar to that with the unconstrained technique (Figure 3A,C). The results of variation partitioning between the model variables showed partial overlap between the snow thickness, mineral nitrogen concentration and water temperature; these factors seemed to have a strong conditional effect on each other (Figure 5B). Nevertheless, every model variable had an individual effect on the explained variation in the range between 5 and 10%: snow thickness, mineral nitrogen concentration, electric conductivity, and water temperature variables yielded 9, 8, 10 and 5%, respectively. These individual effects summed up to 32% of the total 49.2% of variation explained by the model. Thus, the remaining 17% could be thought of as a shared fraction of variation, where the effect of these four explanatory variables could not be disentangled.

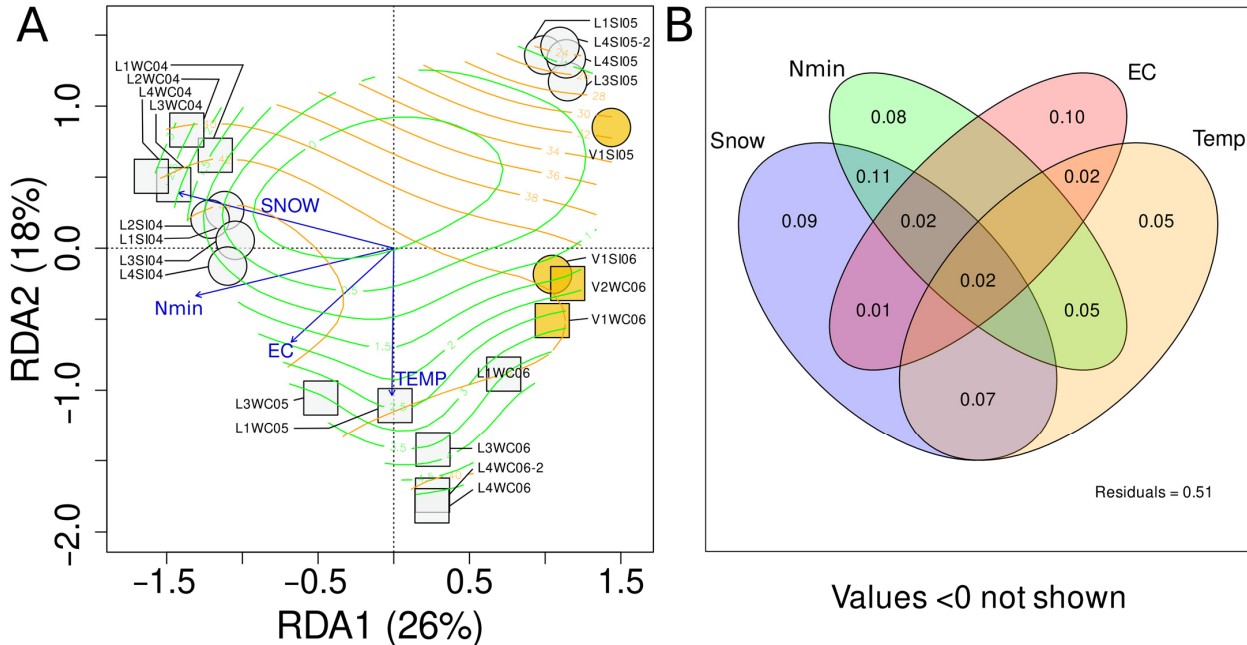

**Figure 5.** Constrained ordination of count matrix with 95 top-abundant OTUs (see Figure 3A,B). (**A**) Ordination plot computed with transformation-based RDA. Exploratory variables were chosen by "forward selection" approach and are indicated by the blue vectors. Point shape: SI—circle, WC—square. Point color: LL—grey, LV—yellow. Gradients of electric conductivity and temperature are drawn by orange and green isolines. Gradients of the first two model variables—Snow and N$_{min}$—are not shown, as the are nearly linear. (**B**) Venn diagram describing the variation partitioning.

### 3.9. Consistency of Results from the Point of View of Technical Replicates

The paired technical replicates, L4SI05/L4SI05-2 and L4WC06/L4WC06-2, could be treated as internal control for consistency of DNA library preparation, amplicon sequencing and data processing framework. The L4WC06/L4WC06-2 and L4SI05/L4SI05-2 were positioned very close on the ordination planes (Figures 3A,C and 5A), had the smallest Bray–Curtis distances (Figure 3B,D), and revealed very similar abundance profiles in the heatmap of major OTUs (Figure 4). Such a robustness was observed in spite of rarefaction curves of L4SI05/L4SI05-2 profiles that were substantially different (Figure 1): the L4SI05 curve was more similar to that of V1SI05 than L4SI05-2. Additionally, the L4SI05-2 curve assumed lower sequencing depth and alpha-diversity indices (Table 2) than that of L4SI05. On the other hand, the rarefaction curves of L4WC06/L4WC06-2 were mainly similar to each other,

as also confirmed by comparable alpha-diversity values. Thus, we argue that the replicates are consistent and expect that our results have no issues from a technical standpoint.

## 4. Discussion

### 4.1. Environmental Factors Influencing the Spatial Differences and Temporal Changes in LL Microeukaryotic Community Composition

Exploratory analysis suggested that during the ice cover period, LL SI and WC profiles are different (Figures 3 and 5). This difference is relatively small in April, while WC and SI profiles are much less similar to each other in May–June. Electric conductivity and temperature were found to be explanatory variables that separated the SI and WC profiles (Table 4, Figure 5). As reverse stratification was expected during the ice cover period [44,45], electric conductivity and, consequently, mineralization in the sub-ice water layer were greater than in the water column [13,46]. This fact can be clearly demonstrated by the difference in EC between LL April samples, when the ice cover is stable (Figure 5A, quadrant IV). In mid May, the ice cover is starting to degrade because of the increase in average air temperature, thinning of the snow cover (Table 2), and the increase in light flux incoming into the under-ice environment. Degradation of the bottom surface of the ice results in demineralization of the sub-ice water layer. This moment is characterized by a sharp drop in the electric conductivity in sub-ice layer, while that of the water column remains constant (Figure 5A, quadrants I and III, Table 2).

Interestingly, an important factor influencing the community structure and revealed by the constrained ordination was the concentration of mineral nitrogen (Figures 5 and S1(D10), Table S1). It was high in the April samples and decreased in May–June. This fact can be explained by the overall increase in ecosystem productivity linked with the increment in light flux [13]. The growth in primary production is associated with the increase in TMA, which was observed in WC May samples (Figure S1(D11)) and resulted in a decrease in free inorganic nutrients within the ecosystem.

### 4.2. Seasonal Dynamics of Microeukaryotic Community Structure

Considering **the** LV June samples, it should be noted that V1SI06/V1WC06 were collected after the process of active ice degradation began. It is not surprising that all three June LV profiles, including V1SI06, were grouped together (Figures 3 and 5). Importantly, the LV May SI profile (V1SI05) and the corresponding LL May SI ones were very similar. Additionally, although good support was found for LL and LV ecosystems to be different in a set of hydrochemical parameters (see results of ANOVA), no clear differentiation of LL/LV microeukaryotic community profiles was observed. Taking this fact into consideration, one may suggest the changes in the microeukaryotic community composition in LL and LV were driven by analogous forces and followed the general compositional trend in May–June.

Summarizing the findings of the differential abundance analysis of LL April microeukaryotic communities, during the ice cover period, these were targeted to degrade the organic matter synthesized in the sub-ice layer. The sub-ice layer community was mainly responsible for primary production mediated by diatoms and dinoflagellates. Bacterivorous flagellates, dinoflagellates, ciliates, and Cercosoa are facultative heterotrophs that are able to compensate for a lack of nutrients and energy by feeding on bacterioplankton [47,48]. The LL WC microeukaryotic community profiles of May–June revealed an increased species richness. On the other hand, the SI May microeukaryotic communities were less diverse and mainly dominated by dinoflagellates. The peak of development of phytoplankton species, such as diatoms and green algae, suggests an increment in primary production at the end of May. Additionally, the primary production was facilitated by photosynthesis of mixotrophic species such as Chrysophyceae and Dinophyceae [35,36]. Considering the taxonomic affiliation of OTUs in LL and LV microeukaryotic communities on a global scale, we found a number of good OTU sequence matches with 18S rRNA barcodes from a wide variety of natural freshwater/brackish, arctic/boreal/temperate

environments in Europe, Asia, North America, and even Antarctica (Table 5). This fact suggests that lacustrine ecosystems around the Earth are not isolated, and the exchange of species between geographically distant sites took place recently or is under way right now [49].

*4.3. Peculiarities of the Specific Microeukaryotic High-Rank Phylotypes in LL and LV*

HTS revealed a number of OTUs assigned to taxonomic groups of microeukaryotes that were never observed in LL and LV using microscopy. This allowed us to extend the knowledge on the species richness and diversity of the insufficiently explored phylotypes. While dinoflagellates, Chrysophyceae, Bacillariophyceae, Chlorophyta, and Prymnesiophyceae were observed in LL/LV by previous studies [10], the Ciliophora-specific OTUs, fungal OTUs assigned to Ascomycota and Basidiomycota, as well as many other eukaryotic phylotypes, were not previously reported in these lakes.

Dinoflagellate phylotypes were constantly present in the under-ice microeukaryotic communities of LL/LV. Their relative abundance decreased in June profiles and was relatively more constant in those of LL than of LV. The closest relatives of the most abundant dinoflagellate-specific OTUs were *Peridinium aciculiferum*/*Scrippsiella hangoei* (OTU3, which was constantly present in all profiles) [29], and *Woloszynskia pascheri* (OTU50, which was more abundant in SI profiles) [30]. These species are the typical members of spring phytoplankton blooms in under-ice freshwater ecosystems [39,40,50]. The massive bloom of dinoflagellates was reported in both marine [51] and freshwater environments [14,15,52], and likely accounts for a considerable proportion of the primary production in LL/LV ecosystems under high snow-cover thickness conditions. For instance, the sub-ice communities of Franklin Bay (Beaufort Sea, Canada) were shown to be dominated by dinoflagellates if the snow mantle thickness was considerable [53]. Furthermore, many dinoflagellate species are adapted to switch to a mixotrophic diet in low light conditions by feeding on bacterioplankton [49,50,54].

The composition of the chrysophyte-specific phylotypes was also changed from April to June. In April, the abundant chrysophyte-specific OTUs were observed in the SI layer; they proliferated in both SI and WC layers in May, while being abundant in WC in June. Chrysophytes are nanoflagellates, which are characterized by a small cell size and motile flagella. Most of the chrysophyte-specific OTUs were revealed to be similar with different species from a variety of freshwater environments (Table S3). The species richness of chrysophytes in LL and LV was studied previously by microscopy. Nineteen species of chrysophytes were classified as *Chrysosphaerella*, *Paraphysomonas*, *Lepidochromonas*, *Spiniferomonas*, and *Mallomonas* genera [9]. Silica-scaled chrysophytes are believed to have an advantage in ice-covered environments by covering a wide spectrum of trophic modes. The major factor influencing the change in abundance of phototrophic and mixotrophic chrysophytes is hypothesized to be the light influx into the under-ice environment [9]. Obviously, the available light influx depends on the snow cover thickness and the ice properties [13]. Notably, we reported the presence of chrysophyte-specific phylotypes (OTU12, OTU444, Table S3), which have high similarity with *Chrysolepidomonas dendrolepidota* from Lake Superior (USA) [31], Spumella-like isolate from Lake Hallstatt (Austria) [32], and *Paraphysomonas foraminifera* from the lake Mondsee (Austria) [32] (Table 5). These phylotypes were not previously detected in LL and LV by microscopy methods.

Ciliophora-specific phylotypes were not previously reported in LL and LV by microscopy. However, a variety of ciliate species were reported to inhabit cold freshwater environments. We detected several Intramacronucleata-specific OTUs, which were classified as Spirotrichea and had the closest sequence hits with SSU of free-living Oligohymenophorea [34], *Cyrtostrombidium longisomum* [35], and *Limnostrombidium viride* SW2012122001 [36] (Table 5). Ciliophora species are known to eventually bloom during the ice-cover period in Lake Baikal, as revealed by microscopy [55] and HTS [56]. Most likely, the seasonal dynamics of ciliate-specific phylotypes result from the change in the nutrient supply [57]. Ciliates are able to effectively feed on phytoplankton during the early

spring blooms, and their diversity is enhanced when the abundance of phytoplankton species changes due to succession [58].

Bacillariophyta were one of the most abundant microeukaryotic phylotypes, which were present in both the SI and WC layers. According to previous study by optical microscopy, diatom species were shown to be quite specific and diverse in LL and LV [6]. Interestingly, more than half of the described species were detected in the under-ice environment. The abundance of diatoms was increased during the ice-cover period [10], with more than 80% of cold-adapted diatom species, such as phytoplankton and periphyton, continuing to grow in the open water conditions in summer [6]. Such a scenario of under-ice proliferation of diatom species is in line with recent reports of studies on phytoplankton in the long-term ice-covered freshwater lakes [14,59–61].

HTS allowed us to detect cercozoan phylotypes in the LL/LV microeukaryotic community profiles. Cercozoa are a diverse group of protists that have no common morphological features required for effective detection by microscopy methods. The Thecofilosea phylotype was abundant in all April microeukaryotic profiles. This taxon of amoeboid protists is known to contain mainly freshwater algivores, which are able to feed on unicellular algae and fungal cells [62]. Thus, Thecofilosea should be thought of as a predator taxon with a wide spectrum of dietary habits.

Fungal-specific phylotypes were not previously reported in LL and LV. At the same time, members of Basidiomycota/Cystobasidiomycetes were detected during all three sampling months, with the peak of abundance in LL in April and LV in June. Cystobasidiomycetes were reported to grow in psychrophylic environments, as demonstrated by the strains isolated from East Ongul Island, East Antarctica [63]. *Cystobasidium tubakii* and *Cystobasidium ongulense*, which are highly similar to OTU7, were reported to grow in below-zero temperatures on a vitamin-free medium. Such features were apparently evolved in these species during their adaptation to the cold and nutrient-depleted environments of subarctic lakes near the Pole of Cold in the northern hemisphere. Ascomycota-specific phylotypes mainly belonged to Thelebolaceae. These phylotypes mainly proliferated in the SI layer, which is also in line with their known features. Several species in the Thelebolaceae family were reported to express antifreeze and ice-binding proteins [64].

Cryptophyte-specific phylotypes were evenly distributed across all of the groups of the microeukaryotic community profiles. This can be explained by the fact that Katablepharidophyta, detected as one of the major OTUs with relative abundance above 0.01%, are characterized as a bacterivore species [65], and this phylotype does not strongly depend on dietary habits. Some Cryptophyte species were reported to be psychrophylic and are sensitive to light conditions. Thus, Chyptophytes are expected to be abundant during the under-ice period with a thick snow mantle. Cryptophytes were reported to proliferate during the whole winter season in high-mountain and subarctic/arctic freshwater environments. Furthermore, Cryptophyte species dominate the phytoplankton in the ice-covered freshwater Antarctic reservoirs, where they represent up to 70% of phytoplankton biomass [66].

## 5. Conclusions

In this study, the first inventory of the microeukaryotic communities of two large subarctic freshwater lakes in Yakutia was compiled by using HTS of 18S rRNA amplicons. As revealed by HTS, dinoflagellates, chrysophytes, Bacillariophyta, green algae, ciliates and Cercosoa were the major classes of phytoplankton and unicellular heterotrophs in the microeukaryotic communities. During the end of the ice cover period, from April to June, the complex communities of microeukaryotes developed in these lakes under harsh environmental conditions characterized by low temperature and decreased inorganic nutrient concentrations. We confirmed that the overall pattern of seasonal changes was shared between the lakes and likely was similar across different years. Importantly, the stratification of the sub-ice layer and the water column communities was constantly

observed during the ice cover period, emphasizing the substantial role of temperature gradient and solar irradiance in the microeukaryotic community differentiation.

**Supplementary Materials:** The following supporting information can be downloaded at: https://www.mdpi.com/article/10.3390/d15030454/s1, Table S1: Impact of exploratory variables evaluated by redundancy analysis and "forward selection" approach; Table S2: Standardized abundance of top-ten classes of LL and LV microeukaryotic communities; Table S3: Taxonomic assignment and the closest GenBank matches of OTU sequences; Table S4: Manually curated taxonomic assignments of OTUs with standardized abundance above 10; Table S5: Differential abundance analysis of LL and LV microeukaryotic communities; Figure S1: One-way ANOVA of environmental variables with categorical variables "Lake", "Month" and "Layer"; Figure S2: One-way ANOVA of alpha-diversity metrics with categorical variables "Lake", "Month" and "Layer".

**Author Contributions:** Conceptualization, Y.L., Y.G. and Y.Z.; methodology, Y.G.; software, Y.G.; validation, Y.G., Y.Z. and D.P.; formal analysis, Y.G.; investigation, Y.G., Y.Z., M.B., L.K. and D.P.; writing—original draft preparation, Y.G.; writing—review and editing, Y.G., Y.Z., Y.L. and M.B.; visualization, Y.G.; supervision, Y.L.; project administration, Y.L. All authors have read and agreed to the published version of the manuscript.

**Funding:** The work was supported by the State Assignments of the Institute for Biological Problems of Cryolithozone, (FWSR-2021-0023, 121012190038-0; expeditions) and of the Limnological Institute (FWSR-2021-0008, 121032300186-9; analysis) of the Siberian Branch of the Russian Academy of Sciences.

**Data Availability Statement:** The sequence data were submitted to the Sequence Read Archive database (https://www.ncbi.nlm.nih.gov/sra, 19 December 2022) of the National Center for Biotechnology Information under Bio Project accession number PRJNA913696.

**Acknowledgments:** The authors acknowledge A.A. Dolzhenkov and I.E. Zhullyarov for organizing the expedition and technical support, and the divers A.S. Gubin, M.V. Astakhov, S.V. Bulochkin, and V.I. Chernykh for their assistance during the field expeditions. We gratefully acknowledge Irkutsk Supercomputer Center of Siberian Branch of the Russian Academy of Sciences for providing the access to HPC-cluster "Akademik V.M. Matrosov" and Shared Equipment Center for Integrated information and computing network of Irkutsk Research and Educational Complex for the data storage infrastructure. We would like to thank Reviewers for taking the time and effort necessary to review the manuscript. We sincerely appreciate all comments and suggestions of Reviewers, especially anonymous Reviewer #1, which helped us to improve the quality of the manuscript.

**Conflicts of Interest:** The authors declare no competing interests.

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
