# Peer review of "Microeukaryotic Communities of the Long-Term Ice-Covered Freshwater Lakes in the Subarctic Region of Yakutia, Russia"

_diversity, doi:10.3390/d15030454_

Round 1

Reviewer 1 Report

Authors have interesting data and did many different analyses but their wording, missing information and incomplete figures and tables hinder a complete understanding. Also the comparison of 4 samples of one lake versus many others from another lake is not the best. In the discussion section, authors write in the present tense, but when writing about their results must use past tense!!!!!!!!!!! The discussion is a mixture of results and discussion and it too long; different months are discussed separately but this destroys the whole picture. There is a hint to lake Baikal in the abstract but further in the discussion I did not find any. The discussion needs a full rewriting to make it more concise.

Abstract:

Please describe in the abstract which analyses were used to analyse your data

Please explain why you are referring to L. Baikal and how the comparison could be achieved

L15: please write: during the end of the ice cover period

L17-18: “The stratification of sub‐ice …. “; it seems that this is an important sentence, but the sentence does not say anything specific; please rewrite in more detail what you are describing

Introduction

L39: please write: were not analysed by metagenomics and high-throughput sequencing (HTS)

L40: please write: “A typical …”; furthermore, please specify what is low temperature (e.g. < 10°C at the surface)

L46:please write: “upper layer of the water column”

L54: “These findings are relevant to compare with results of the current” -> please rephrase

L57-63: this paragraph needs a rewriting regarding the style: please use something like: we compiled, we evaluated, we discussed

L58: Instead of the metabarcoding approach, write metabarcoding of 18sRNA.

Materials and Methods

L66: of the geographical …

L69: not QC but quality control

L71: please explain why you used this prefiltering with a 27 µm mesh

Legends for Table 1 and 2 are missing: this is unfortunate; nevertheless, I was expecting to find the same sample names in table 1 and 2 what is not the case

L109: was instead of is

L128: when authors do not know the exact name of their analysis, this is always suspicious; in any case; tb‐PCoA is Principal Coordinates Analysis

L138: I suggest using the Hellinger transformation instead of the log transformation this is the recommended transformation for abundance data and RDA (Ecologically Meaningful Transformations for Ordination of Species Data Pierre Legendre and Eugene D. Gallagher)

L147: it is completely unknown why there is a upper case and lower case April, May,..; please clarify their meaning

Also this is unfortunate that I cannot see supplementary material….

With huge amount of statistics on HTS data, I am wondering why you did not some statistics on environmental data? Please do so, to combine this with your HTS analyses

Results

L158-159: this sentence has to be moved to the method section; by the way, please clarify from which depth to where did you sample the water column; furthermore, what spatial extension is the sub-ice layer (e.g. 0.5 m above the ice?)?

L163-167: this would be the right place to state differences between layers and lakes by performing ANOVA

L169: because it is completely unclear from where this sample (Sample V1WC05) is coming, you can put this info in the method section and clarify from where (layer and lake) it is

L171: “remaining” instead of “rest”

L172: L4WC06 and L4SI05; it is completely unknown how you coded your samples; finally, I understood that WC is for water column and SI is for under ice; nevertheless, what are the other letters and numbers, please clarify!

Figure 1: sample names are impossible to read; please make them larger and explain your coding

L176: I did not find any trace of rarefying the OTU table to the lowest abundance obtained for a sample; did you do it? If yes, clarify; if not, then clarify why not

L182: “increased” instead of “increase”

Figure 2 does not have any legend and it covers text ….; you cannot refer to a figure that comes later than figure 1; in any case, I suggest separating SI and WC samples (make two plots for them with one below another) for a better temporal comparison; while now I understand that numbers refer to months, still I do not understand what L3, L4, … is; in any case, because you also have tow lakes, separate the two lakes from each other -> this means 4 plots in total!

Figure 2 is an important figure but in the present form it is useless

L188: The generated …

L189: included… represented

L193-200: the present figure makes it impossible to see/control this

L220: this is the tb-PCoA, please state it like that! What are orphans? Please clarify how you got the arrows for environmental variables!

L225: based on which metric did you infer that it was the least similar?

Figure 3: please clarify what are orphans;

I did not find any clear description of the clustering result…

The heatmap is quite colorful and quite complex; honestly, I do not see any clear pattern; instead, I would prefer to see the analysis on differential abundance that gives clear results and does not depend on the reader`s ability to see patterns; therefore, I suggest moving this analysis to the main text

L228: how did you come to the statement of most diverse? Both analyses indicate differences but do not tell anything which group is most diverse. Please, pay more attention to your wording.

L232-233: you are referring two time to the same samples; this is confusing

L227-241: I have serious problems following your reasoning; please rephrase or provide a table where it is easier to see these patterns

L252: what is moderate abundance? What is increased abundance? Please clarify

L269: total explained variance for the first two axis, right?

Figure 5: I see that you liked playing with vegan; in any case, either a variable has a linear dependence, or it is non-linear but you are showing both for temperature and conductivity; I could accept this, if you could explain it better in the text; presently, one has to know quite well you analyses to understand what and how you did it

L270-275: I would like to see which variable has the highest independent contribution, which the lowest and how large is the interaction between all of them

Discussion

L282-294: move to result section, this has nothing to do with discussion

L296: “correctly”? HTS has so many sequencing biases that there is no way to know if the picture you got is the correct on; please delete the sentence or rephrase

L302: it is not NGS but HTS

L295-305: this section seems to be the awkward attempt to tackle NGS biases versus how cool your data are; please be more concise by writing about primer bias, sequencing depth, ecc. and acknowledge that another primer might give other results; furthermore, I agree that your data are sound and you can infer patterns from them

L314: you write about centroids without showing them; in fact, in vegan it is possible to draw the 95% confidence ellipses of the group centroid (doing this would help your statements)

L320: this statement needs a citation!

L320-322: strange statement; please, rephrase

L333-335: result and not discussion; I would like to see more of such clear and precise statements in the result section

L336-363: I only read results, interesting ones but no discussion at all; several typos in this section

L372-377: is the only discussion in this section; typos

L375-377: needs citation

L378: the section with this title is more a discussion than the previous paragraph but nevertheless is quite long; I suggest putting some text to the result section any focusing more on the discussion part

I read the whole discussion section that mixes results with discussion and is quite long; I was remembering some hint at a comparison with Lake Baikal but at the end there was none

Reviewer 2 Report

Please see the atatchment.

Reviewer 3 Report

This work describes the microeukaryotic community in two arctic lakes and tries to find factors affecting the community structures. Data were well analysis and visualization. It is a good job and I think readers will be interested.  However, there are several points the authors have to be revised, especially in the discussion section.

Major comments:

In the discussion, the authors address the specific OTUs too much which is not the aim of this manuscript. I suggest the author should focus and carefully think about the reason why there are differences between two lakes and seasons because, in the abstract, the primary purpose is to find the reason why they are different or the same.

minor comments

1. Table 1, add a unit for each parameter.

2.     Usearch and Vsearch both have the same function. Which one did author actually use? If use both, please provide the reason in the methods section.

3.     Result, move the first sentence to the methods section.

4.     Is the totN total nitrogen?

5.     Line 346, species

6.     Line 346, SSU rRNA gene sequences

7.     Line 372-377, the conclusion is weak and literature is needed here. And I cannot understand how it is concluded.

8.     Line 415, reword the sentence.

9.     English also need to be polished.

Author Response

This Review is the same as #2. So I am attaching our response to Reviewer #2.

Round 2

Reviewer 1 Report

I congratulate authors to their effort to improve their manuscript, well done. Nevertheless, please find some more issues that should be addressed to further improve this study.

Abstract

L19: please write …. communities differentiated …

Introduction

L42: just write …LL and LV are ice covered …

L63: insert space before “Over”

Material & Methods

Table 1: IMPORTANT: while the legend now states in detail what column Sample shows, I do not see LL or LV as described in the text!!! Please check again your labelling of samples in the first column [arriving at Figure 2, I finally understood: L is LL and V is LV; please state it accordingly in tables and figures]

Based on table 1 I deduce that you sampled 4 times during April and 3 times during May for LL and once for LV and 5 times for LV during June. Is this correct? Please give detailed information on the sampling time in the text!

L157: for “the raw community composition matrix was standardized by …” please provide a citation that explains how this is done and why

L181: please write FDR in full because this is an abbreviation.

Results

L248-249: past tense

L251: what do you mean with “possessed the robust temporal dependence”? The sense of doing an ANOVA is to inspect for differences between factors (here month); once you found differences by a significant ANOVA result, you perform post-hoc testing to see which levels of the factor (here the different months) are different between each other; you then write it in the text, was this would be the correct procedure

L254: please write: … TMA  gradually increased from April to June

L256-258: please write: However, the only variable different between layers was water temperature that obviously was lower in the SI layer than the WC layer.

L262: please write: …. differed significantly over time ….

L277-280: past tense

L301: please write: revealed that

L303: please write: in tc-PCoA, LL WC microeukaryotic communities of May‐June formed slightly diffuse group

L303-308: honestly, I do not like how you describe your results (outgroup?, diffuse group?), BUT the good thing is that figure 3 is so clear that readers will see the month and layer differences. My recommendation would be to describe the 5 groups that are there: April WC similar to April SI; SI May completely different, forming a distinct group; WC and SI June completely different; WC May forming a distinct group.

L309: write “showed that” instead of “suggested”

L316: please specify what profiles you are referring to, WC or SI?

L161+313: based on what metric is the heatmap drawn? Correlation? Please specify this info in the methods section

L334: relative

L336: this table 4 is not agreeing with table 4 shown in the text

L337: please write: the following OTS were abundant in LL SI profiles: ….

L338-340: past tense

L337-424: I agree with you that you have to list all OTUs from different seasons and layers that where most abundant and unique! However, the text is quite long and I suggest that you summarize this information in a table by stating which OTU at which similarity level is most similar to + the citation; this would give a better overview and be much shorter

L444-449: please write: In tb-PCA, variability explained of community composition was 67.5 %; selecting for the most important environmental predictors with forward selection, variability explained was 49.2% and variables selected were….

L458: please delete (“see my answer above”)

L458-459: please check https://r.qcbs.ca/workshop10/book-en/variation-partitioning.html and rephrase the sentence on interaction

Discussion

L479-490: move the whole paragraph to the result, either as the first or the last paragraph

L499: HTS instead of NGS

L491-494: these technical replicates are proofs how well you can repeat your results, and you did well, congratulations; however, these are not proofs that you sufficiently covered the whole community, just think of a primer bias! By constructing a mock community, you could test for primer bias and sequencing efficiency. In any case, there is no need to argue on this topic, just write that technical replicates were fine and therefore no issues with analyses are expected BUT do not write of the complete coverage of the protist community.

L510-515: rephrase and shorten the text because it is too long and repetitive and strange wording

L537: delete “and”? ; the sentence is difficult to understand

L547: delete “in its turn”

L588: rephrase “were actively proliferate”, impossible to understand

L590: “were” instead of “was”

L616: typo for Bacillariophyta

L617-619: split the sentence in two because you are mixing you results with the results of others

L632: delete “in”

Reviewer 3 Report

Authors revised the manuscript very carefully. The present version is acceptable. Congratulations!
